theoretical biology/mathematical modelling/
biomathematics

microbiome, parameter identifiability,
mathematical model, generalized Lotka–Volterra,
time-series, dynamics

**Author for correspondence:**
Christopher H. Remien
e-mail: cremien@uidaho.edu

# Structural identifiability of the generalized Lotka–Volterra model for microbiome studies

Christopher H. Remien[1], Mariah J. Eckwright[2] and Benjamin J. Ridenhour[1]

[1]Department of Mathematics and Statistical Science, and [2]Bioinformatics and Computational Biology Program, University of Idaho, Moscow, ID, USA

CHR, 0000-0003-1179-9041; BJR, 0000-0001-8271-4629

Population dynamic models can be used in conjunction with time series of species abundances to infer interactions. Understanding microbial interactions is a prerequisite for numerous goals in microbiome research, including predicting how populations change over time, determining how manipulations of microbiomes affect dynamics and designing synthetic microbiomes to perform tasks. As such, there is great interest in adapting population dynamic theory for microbial systems. Despite the appeal, numerous hurdles exist. One hurdle is that the data commonly obtained from DNA sequencing yield estimates of relative abundances, while population dynamic models such as the generalized Lotka–Volterra model track absolute abundances or densities. It is not clear whether relative abundance data alone can be used to infer parameters of population dynamic models such as the Lotka–Volterra model. We used structural identifiability analyses to determine the extent to which a time series of relative abundances can be used to parametrize the generalized Lotka–Volterra model. We found that only with absolute abundance data to accompany relative abundance estimates from sequencing can all parameters be uniquely identified. However, relative abundance data alone do contain information on relative interaction strengths, which is sufficient for many studies where the goal is to estimate key interactions and their effects on dynamics. Using synthetic data of a simple community for which we know the underlying structure, local practical identifiability analysis showed that modest amounts of both process and measurement error do not fundamentally affect these identifiability properties.

# 1. Introduction

There is considerable interest in applying population dynamic theory to microbial systems to test hypotheses relating to ecosystem stability, to determine the drivers of dynamics and to predict how populations will change over time (e.g. to prevent illnesses such as ulcers; cf. [1–9]). While modern DNA sequencing technologies allow rapid and inexpensive characterization of microbial community composition and have uncovered enormous microbial diversity, relatively less is known regarding the interactions governing the population dynamics of constituent members of microbial communities. It is these interactions that determine which members of a microbial community will flourish, and understanding them is key to manipulating microbiomes to promote health, designing synthetic microbial communities to perform tasks and inferring stability to assess risk.

Despite the expectation that population dynamic models should be applicable to microbial systems, barriers exist to the application of traditional modelling approaches to microbiomes. One such barrier results from the nature of sequence data, which are used as a proxy for species abundance. The raw data from high-throughput microbiome samples are a large number of sequence reads which are grouped by similarity, giving the number of reads belonging to a particular group. These groups have different meanings depending on the methods employed (e.g. unique sequence variant, operational taxonomic unit, species) but, for our purposes, can be thought of as different (pseudo-)species. The number of reads for a group is then divided by the total number of sequence reads in the sample giving an estimate of relative abundance. In contrast to the relative abundance estimates obtained from sequence reads, most population dynamic models, including the generalized Lotka–Volterra (gLV) model, describe absolute abundances or densities rather than sequence observation rates or relative abundances. Methods exist to convert the relative abundance data to absolute abundances by estimating absolute abundance from additional data (e.g. qPCR) [8,10,11]. However, such data are not typically collected in microbiome studies and, when collected, are quite error-prone themselves.

Numerous methods exist to estimate species' interaction strengths in a gLV model from microbial time-series data [8,10–15]. The most common technique for estimating parameters is to use a discrete-time version of the model, and estimate coefficients using gradient matching fit with linear regression. When formulated in this way, it has been recognized that the design matrix for the regression is singular [13], because relative abundance data alone do not contain sufficient information to estimate parameters. As such, methods typically rely on an assumption of constant population size or additional data on absolute abundance to complement sequence data so that absolute abundances or densities of each species can be estimated. While interaction strengths have been successfully estimated in microbial communities by fitting time-series of species' densities to the gLV model, it is unknown to what extent relative abundances alone contain information on interaction strengths.

More broadly, the question of how well experimental data can be used to estimate parameters of a mathematical model can be addressed with parameter identifiability analysis [16–18]. There are two broad categories of parameter identifiability analysis: structural and practical identifiability. Structural identifiability is a global property of a model and measurement type. It addresses the extent to which parameters of the model can be estimated assuming that all observations are error free. Structural identifiability can be assessed before collecting experimental data. By contrast, the related concept of practical identifiability depends explicitly on the quality and quantity of experimental data and is a property of the model and observation type alone. Practical identifiability, which can be defined globally over the entire parameter space or locally near a critical point of interest, addresses the extent to which parameters of the model can be estimated given a set of experimental data. Practical non-identifiability may arise, for example, from poor data quality even when such parameters could be identifiable with better quality data.

Here, we first determine the extent to which time-series of microbiome sequencing data contain information about parameters of the gLV model using structural parameter identifiability analyses to systematically account for the compositional nature of the data. We address the question as to whether relative abundance measurements alone, as obtained by sequencing techniques, can be used to estimate species interaction strengths in a gLV model. If relative abundance measurements cannot be used to estimate all species interaction strengths as has been previously suggested, what additional measurements would be needed to make this possible, and what parameters or combinations of parameters can be estimated using only relative abundance measurements? We then verify the structural identifiability results by studying local practical identifiability with synthetic microbial community time-series data.

# 2. Analysis

## 2.1. Generalized Lotka–Volterra model

Assuming large, well-mixed, closed populations with only two-way interactions between microbes, the change in density of microbes over time can be described by a system of differential equations where the dynamics of a focal microbe $N_i$ satisfy

$$\frac{\mathrm{d}N_i}{\mathrm{d}t} = \underbrace{h_i(N_i)}_{\text{intrinsic growth or death of } N_i} + \underbrace{\sum_{j=1}^{n} f_{ij}(N_i, N_j)}_{\substack{\text{growth or death of } N_i \text{ caused by} \\ \text{interaction with microbe } N_j}} , \tag{2.1}$$

for $i \in \{1, 2, \ldots, n\}$, where $n$ is the number of species in the microbial community. The function $h_i(N_i)$ is the rate of growth or death of $N_i$ and $f_{ij}(N_i, N_j)$ is a function describing the growth or death of $N_i$ caused by interaction with microbe $N_j$. Growth or death of $N_i$ caused by interaction with exogenous variables (e.g. resource, toxin) can be added in a similar manner. Specifying the functions $h_i(N_i) = r_i N_i$ and $f_{ij}(N_i, N_j) = \beta_{i,j} N_i N_j$, and dividing by $N_i$ yields the classical generalized Lotka–Volterra model

$$\frac{1}{N_i} \frac{\mathrm{d}N_i}{\mathrm{d}t} = r_i + \sum_{j=1}^{n} \beta_{i,j} N_j, \tag{2.2}$$

where the parameter $r_i$ is a positive growth rate and the interaction rate $\beta_{i,j}$ describes how microbe $j$ affects the growth rate of microbe $i$. Typically, the parameters $\beta_{i,i}$ are constrained to be negative so that the carrying capacity $K_i = -r_i/\beta_{i,i}$ is positive, but this is not required for the structural identifiability analysis.

## 2.2. Structural identifiability of generalized Lotka–Volterra with relative abundance data

We wish to determine the upper bound on the information contained in a time series of relative abundance data in relation to the gLV model defined in equation (2.2). As previously mentioned, after grouping by similarity, sequencing data give estimates of the relative abundance of each group, whereas equation (2.2) tracks absolute abundances or densities. In this section, we will use existing methods for structural identifiability analyses to determine the extent to which relative abundance data can be used to infer population dynamic parameters.

A given model and observation state combination is said to be 'structurally identifiable' if it is possible to uniquely estimate the parameters of the model assuming error-free measurements [19,20]. The goal of structural identifiability analyses is to identify model parameters that cannot be estimated from a given data type. Moreover, analysis of structural identifiability can reveal parameters or combinations of parameters that are uniquely identifiable, and can inform the reparametrization of a model in terms of identifiable combinations of parameters. For the gLV model in equation (2.2), the structural identifiability problem can be set up as follows: given a noise-free time series of the relative abundance of each microbe (i.e. $N_i/N$ for all $i$ where $N = \sum_i N_i$), determine whether it is possible to estimate parameters $r_i$ and $\beta_{i,j}$ for all $i$ and $j$ of equation (2.2).

A variety of methods exist to determine structural identifiability of ODE models [16]. We will use a differential algebraic approach which is relevant for rational-function ODE models such as equation (2.2). For more details on this method, we refer the reader to Saccomani *et al.* [20], Audoly *et al.* [19] and Eisenberg *et al.* [21]. To apply this method to the model described in equation (2.2), the idea is to first algebraically manipulate the system of differential equations into an equivalent system written only in terms of *observable* state variables (i.e. the measured data) and their derivatives. The resulting system can be regarded as a system of differential algebraic equations (DAEs) with polynomial coefficients which, after dividing by the coefficient of the highest ranking polynomial to make the resulting system monic, leads to an input–output relation that has identifiable coefficients [21,22]. While this method is theoretically valid for a microbial community of arbitrary size, the algebra becomes cumbersome for even relatively small communities. Nevertheless, analyses of communities of small size uncover clearly recognizable patterns that appear to be broadly applicable to communities of arbitrary size.

We begin by analysing the structural identifiability of the two-species gLV model

$$\left.\begin{array}{l} X' = r_1 X + \beta_{1,1} X^2 + \beta_{1,2} XY, \\[4pt] Y' = r_2 Y + \beta_{2,1} XY + \beta_{2,2} Y^2 \\[4pt] N' = X' + Y', \end{array}\right\} \tag{2.3}$$

and

where $X$ represents the absolute abundance of the first species, $Y$ the absolute abundance of the second species, $N = X + Y$ the total abundance of microbes in the community, and the prime symbol indicates derivatives with respect to time. Note that $N$, $X$ and $Y$ are all time-dependent (e.g. $N(t)$), but we are suppressing the time notation for brevity. Ideally, we would like to use measurements of relative abundances to estimate parameters of (2.3). To check identifiability, we rewrite equations (2.3) in terms of the measurable quantities (relative abundances). Let $x = X/N$ and $y = Y/N$ be relative abundance of microbes $X$ and $Y$, respectively. Differentiating $x$ with respect to $t$ yields

$$x' = X'N^{-1} - XN^{-2}(X' + Y'). \tag{2.4}$$

Utilizing $X = Nx$ and $Y = N(1 - x)$ and equation (2.4), equations (2.3) can be rewritten in terms of relative abundances and $N$ as

$$\left.\begin{aligned}
x' &= x(1 - x)N\left(\frac{(r_1 - r_2)}{N} + (\beta_{1,1} - \beta_{1,2})x + (\beta_{2,2} - \beta_{2,1})(1 - x) + (\beta_{1,2} - \beta_{2,1})\right), \\
y' &= -x'
\end{aligned}\right\} \tag{2.5}$$

and $\quad N' = N(-r_2(x - 1) + r_1 x + N(\beta_{2,2}(x - 1)^2 + x(\beta_{1,2} + \beta_{2,1} - (-\beta_{1,1} + \beta_{1,2} + \beta_{2,1})x))).$

Solving the first equation in (2.5) for $N$ yields

$$N = \frac{(r_1 - r_2)(x - 1)x + x'}{(x - 1)x(\beta_{2,2} + \beta_{1,2}(x - 1) - (\beta_{1,1} - \beta_{2,1} + \beta_{2,2})x)}. \tag{2.6}$$

Substituting equation (2.6) and its derivative into the $N'$ equation in equations (2.5) and collecting terms yields

$$\begin{aligned}
0 = &\, x''(x^3((\beta_{2,1} - \beta_{2,2}) - (\beta_{1,1} - \beta_{1,2})) \\
&\quad - x^2((\beta_{2,1} - 2\beta_{2,2}) - (\beta_{1,1} - 2\beta_{1,2})) \\
&\quad - x(\beta_{2,2} - \beta_{1,2})) \\
&- x'^2(2x^2((\beta_{2,1} - \beta_{2,2}) - (\beta_{1,1} - \beta_{1,2})) \\
&\quad - x((\beta_{2,1} - 2\beta_{2,2}) - (2\beta_{1,1} - 3\beta_{1,2})) + \beta_{1,2}) \\
&- x'(x^3(r_1\beta_{1,1} + r_2\beta_{2,2} + (r_1(\beta_{2,1} - 2\beta_{2,2}) - r_2(2\beta_{1,1} - \beta_{1,2}))) \\
&\quad - x^2(r_1\beta_{1,1} + 2r_2\beta_{2,2} + (r_1(\beta_{2,1} - 4\beta_{2,2}) - 2r_2(\beta_{1,1} - \beta_{1,2}))) \\
&\quad - x(2r_1\beta_{2,2} - r_2(\beta_{1,2} + \beta_{2,2}))) \\
&- x^5(r_1 - r_2)(r_1(\beta_{2,1} - \beta_{2,2}) - r_2(\beta_{1,1} - \beta_{1,2})) \\
&+ x^4(r_1 - r_2)(r_1(2\beta_{2,1} - 3\beta_{2,2}) - r_2(2\beta_{1,1} - 3\beta_{1,2})) \\
&- x^3(r_1 - r_2)(r_1(\beta_{2,1} - 3\beta_{2,2}) - r_2(\beta_{1,1} - 3\beta_{1,2})) \\
&- x^2(r_1 - r_2)(\beta_{2,2}r_1 - \beta_{1,2}r_2). \tag{2.7}
\end{aligned}$$

Dividing equation (2.7) by the leading-order coefficient $((\beta_{2,1} - \beta_{2,2}) - (\beta_{1,1} - \beta_{1,2}))$ produces an input–output DAE strictly in terms of $x''$, $x'$ and $x$ whose coefficients are identifiable. Finally, we check whether coefficients of the DAE have a unique solution by considering an alternative set of parameters $(a_1, a_2, a_3, a_4, a_5, a_6)$ that produces the same output. Doing so gives the following result: $r_1 = a_1$, $r_2 = a_2$, $\beta_{1,1} = (a_3/a_6)\beta_{2,2}$, $\beta_{1,2} = (a_4/a_6)\beta_{2,2}$ and $\beta_{2,1} = (a_5/a_6)\beta_{2,2}$. Thus, $r_1$ and $r_2$ are identifiable using relative abundance data, while $\beta_{i,j}$ are not identifiable; however, $\beta_{i,j}$ are identifiable up to a constant. Reparametrizing model (2.3) in terms of carrying capacities does not fundamentally change results; the growth rates are identifiable while interactions and carrying capacities are identifiable up to a scaling factor.

We performed similar analyses for three-, four- and five-species gLV models and found similar identifiability results (electronic supplementary material, Mathematica Notebook). Parameters $r_i$ are identifiable, while $\beta_{i,j}$ are only identifiable up to a scaling factor. We conjecture that this is true for communities of arbitrary size.

## 2.3. Local practical identifiability in a synthetic community

Here, we use synthetic data to illustrate the structural identifiability results and to test whether modest amounts of process and measurement error affect local practical identifiability. We created two simple three-species synthetic communities that, according to the structural identifiability analysis in the

**Table 1.** Parameter values used in numerical simulations.

| parameter | value 1 | value 2 |
| --- | --- | --- |
| $r_1$ | 6 | 6 |
| $r_2$ | 4 | 4 |
| $r_3$ | 2 | 2 |
| $\beta_{1,1}$ | −0.05 | −5 |
| $\beta_{1,2}$ | 0.15 | 15 |
| $\beta_{1,3}$ | −0.20 | −20 |
| $\beta_{2,1}$ | −0.01 | −1.0 |
| $\beta_{2,2}$ | $-0.02\overline{6}$ | $-2.\overline{6}$ |
| $\beta_{2,3}$ | 0.05 | 5.0 |
| $\beta_{3,1}$ | 0.10 | 10 |
| $\beta_{3,2}$ | −0.10 | −10 |
| $\beta_{3,3}$ | $-0.01\overline{48}$ | $-1.\overline{481}$ |
| $X_1(0)$ | 10 | 0.10 |
| $X_2(0)$ | 14 | 0.14 |
| $X_3(0)$ | 4 | 0.04 |
| $N(0)$ | 28 | 0.28 |

previous section, should yield identical relative abundance time series (table 1). We numerically solved equations (2.3) with parameters in table 1 using the function ode with the default lsoda method in the deSolve R package. As suggested by the structural identifiability results, both sets of parameter values yield identical relative abundance time series (figure 1).

We first created synthetic data with error and then sampled the likelihood surface near the true parameter values to check whether the likelihood surface has a maximum near the true parameter values. Process error was added to the synthetic data using stochastic differential equations. We used a Wiener process (Brownian motion) to model environmental noise to the system [23]

$$dN_i = N_i \left( r_i + \sum_{j=1}^{n} \beta_{i,j} N_j \right) dt + \sigma_i \sqrt{N_i} dW_i, \tag{2.8}$$

where $\sigma_i$ scales the variance of the Wiener process ($dW_i$), which is $\mathcal{N}(0, \sqrt{dt})$-distributed random noise. After addition of the process error, we subsequently added measurement error to the stochastically modelled relative abundances to simulate the aggregation of sequence reads to relative abundance. To do this, we drew random proportions from a Dirichlet distribution having concentration parameters $\alpha_i = V\tilde{N}_i / \sum_i \tilde{N}_i$, where the tilde indicates the population sizes simulated by equation (2.8), and $V$ scales the error magnitude ($V$ is roughly equivalent to the amplicon read count). Simulation of data with process and observation error was performed using the pomp function in the POMP R package [24]. Simulated relative abundance data were generated for parameter values 2 (table 1) with $\sigma_i = 0.1$ for $i = 1, 2, 3$ and $V = 500$.

Next, we checked local practical identifiability near the correct parameter values. We are interested in determining whether a likelihood ridge exists such that the solution could travel outside the basin of attraction of the true parameter set. To sample the likelihood surface, we used POMP's particle Markov chain Monte Carlo (pMCMC) algorithm (cf. [25]), starting the algorithm near the correct parameter values. We tested local practical identifiability for three scenarios: two scenarios with initial population given in table 1 and a third scenario where the initial population size was $N_0 = 1$. The third scenario was performed to demonstrate that *relative* changes in population size can be inferred from relative abundance data (i.e. $N(t) \approx N_0 \hat{N}(t)$, where the hat denotes the estimated population size); that is, the population size relative to initial population size can be estimated. The fitting was performed following the suggested protocol for the POMP software [24]. We assumed that every sample had $V = 500$ amplicon read counts that were then divided with respect to the relative

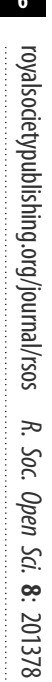

**Figure 1.** Multiple parameter sets lead to identical relative abundance dynamics. Comparison of two gLV systems with different parameters (table 1) that yield different absolute abundances (top panels) but identical relative abundances (bottom panels). The non-identifiable parameters can be scaled by a constant to produce infinitely many systems with identical dynamics in terms of the relative abundances. (The scaling constant chosen here was 100.)

proportions of each species; accordingly, we specified that read counts (measurements) were multinomially distributed. The time-step (d$t$) for the random process error was 0.001. A prior uniform distribution was placed on each parameter, $\rho$, such that the likelihood surface was defined on $\mathcal{U}(\rho - 0.4|\rho|, \rho + 0.4|\rho|)$. The MCMC algorithm was first run for 2000 iterations with 200 particles used for filtering at each iteration. During these iterations, proposals were drawn using a multivariate-normal, adaptive, random walk where the covariance matrix of MVN was defined as a diagonal matrix with non-zero elements corresponding to $\sqrt{0.1} \times |\rho|$; after 100 iterations, a scaled empirical covariance matrix based on the accepted proposal was used. Once the first 2000 iterations finished, the MCMC sampler was restarted and run for another 2000 iterations using the empirically determined covariance matrix from the previous 2000 iterations. After sampling was completed, the final 2000 iterations were thinned by keeping only every 50th sample, resulting in 40 proposals from which to estimate the summary statistics for each parameter.

Testing local practical identifiability using parameters from the two scenarios in table 1 indicates that modest amounts of observation and process error do not fundamentally affect the identifiability properties of the gLV model with relative abundance data in this synthetic community (figure 2; electronic supplementary material: R Markdown File). Because the simulated data were generated with process and observation error, there are deviations between the model fits and the simulated data. Yet the estimated parameters stay within a basin of attraction near the true parameters. Using only relative abundance data and an initial population size, most parameter estimates were within approximately 20% of their true value. Surprisingly, testing local practical identifiability using an initial population size of one indicates that the relative interaction rates—that can be estimated using relative abundance data—provide ample information to estimate relative changes of absolute abundance over

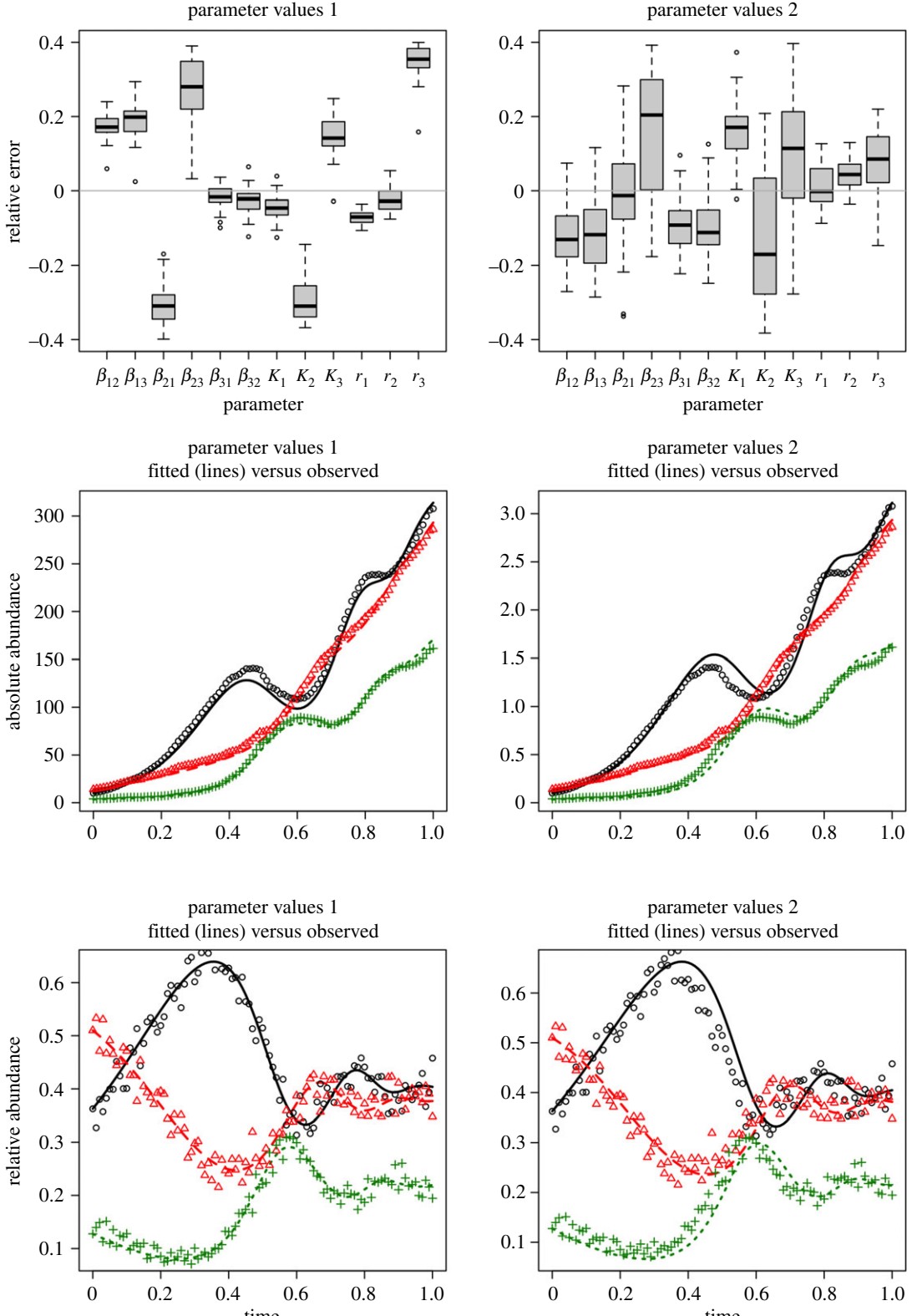

**Figure 2.** Fits of the synthetic three-species gLV system with process and observation error. The left-hand column shows the results for the 'value 1' set of parameters (table 1), while the right-hand column shows the results for the 'value 2' set of parameters. Note that the scales for absolute abundance vary for the absolute abundances generated from the two sets of parameters. Fitting was done by starting the algorithm near the correct parameters set and giving it the correct total population size at $t = 0$ ($N_0 = 0.28$ and $N_0 = 28$, respectively). The fits for the relative abundance data are quite similar (last row), as are the extrapolated fits of the absolute abundance data (middle row). Relative error of the parameter estimates was defined as relative error $= x/\hat{x} - 1$, where $x$ is the estimated value and $\hat{x}$ is the true value. The relative error was bounded on $[-0.4, 0.4]$ by the assumed priors.

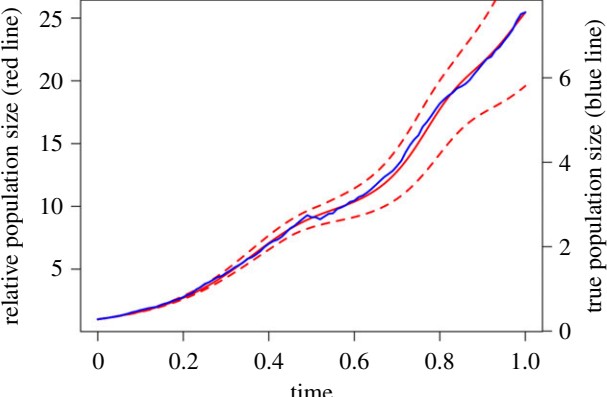

**Figure 3.** Reconstruction of relative population size. If a model is fit where it is assumed that $N_0 = 1$, then information is recovered about the fold-change in the population's size over time. The mean estimate (solid red line) does a fairly good job of matching the stochastic population trajectory (blue line). The dashed red lines show the range of population values produced by numerically solving the three-species gLV system for all accepted parameter sets in the MCMC algorithm.

time (figure 3). Put differently, total abundance relative to initial total abundance can be reasonably estimated from a time series of relative abundances of all microbes in conjunction with a gLV model. Of course, if the model is misspecified, or data sufficiently sparse, such estimation may not be possible for a given study system.

## 3. Discussion

As a simple and general model that describes how interactions shape population dynamics of communities, the gLV is a natural candidate model for interpreting microbial time-series data [5,6,13,26]. To date, fitting the gLV model to time series requires either (i) assuming a constant population size so that the gLV model can be fit directly to relative abundance data; or (ii) performing additional measurements to obtain absolute abundances. We find that such assumptions or additional data are not strictly necessary. Using structural identifiability analysis and numerical simulations, we have shown that relative abundance data alone contain sufficient information to obtain relative rates of interaction. The ability to estimate the topology of an interaction network with only relative abundance time-series data would greatly expand the range of datasets available to interpret with dynamic models, as estimates of absolute abundances are typically unavailable. In many studies, estimating relative interaction rates may be sufficient, as it would still allow for the identification of the key microbial interactions that provide services and drive dynamics. Moreover, if additional information on absolute abundance can be obtained (e.g. through qPCR or optical density measurements) to complement relative abundance data, our results indicate that the parameters of a gLV model can be uniquely identified. Such additional information need not be obtained at each time point; even one measurement of absolute abundance can, in theory, be used as the key piece of information to anchor the parameters giving identifiable estimates.

Just because the relative interaction rates are structurally identifiable with relative abundance data does not mean that they are practically identifiable for all systems, nor does it mean that estimating such parameters will be straightforward. The large number of parameters in systems with many species may preclude accurate parameter estimation in the absence of prior information on parameter values. Even for smaller communities, some parameters may not be practically identifiable due to the nature of the specific system. For example, if a species never gets near its carrying capacity, the carrying capacity may not be well estimated. Similarly, the number of data points is not the only determinant of the amount of information contained in a time series. Time series in which populations exhibit large changes in population sizes due to, for example, perturbations, typically contain more information regarding interactions than time-series of species at steady state. Our local practical identifiability analysis of a small system shows that in principle modest amounts of noise do not preclude parameter estimates. Yet in general, the level and type of noise also clearly dictates the ability to accurately estimate interaction parameters. For example, poor sequencing depth can increase measurement error and yield poor estimates of parameters. At best, fitting data from a large system

that is noisy and sparsely sampled would lead to poor fits and large confidence intervals. At worst, it could lead to a good fit of an entirely incorrect set of parameters and incorrect biological conclusions. As with any statistical method, care must be taken when estimating parameters of the gLV model.

Developing a better understanding of how microbes interact with each other and their environment is required for numerous goals in microbiome research including detecting dysbioses, manipulating microbiomes to promote healthy function and preventing disease, and designing synthetic microbial communities for specific tasks. Such an understanding can be facilitated by interpreting data in conjunction with appropriate mathematical models. Some microbial communities may be better modelled with more complex formulations than the gLV model that incorporate additional factors, for example, higher-order interactions, indirect resource-mediated interactions, time-varying interactions, and various forms of stochasticity. The optimal level of detail to be included in such a model probably depends on numerous factors including the complexity of the microbial community, the level of understanding of the underlying dynamics, the structure of noise in the data and the goals of the study.

Regardless of the underlying model structure, population dynamic models must be adapted to use common forms of data, such as the relative abundance data obtained from high-throughput sequencing. We have found that when fit to a common population dynamic model, the generalized Lotka–Volterra model, a time series of relative abundance data contains information on relative interaction strengths. Moreover, relative interaction rates provide ample information to estimate relative changes of absolute abundance over time. Such findings provide critical information for designing temporal studies aimed at inferring microbial interaction networks, and greatly expand the number of studies amenable to such analysis. Specifically, we have shown that qPCR data—to convert relative abundance into absolute density—are not strictly necessary to obtain such networks. By appropriately connecting mechanistic models like the gLV with relative abundance data, we can potentially tease apart meaningful interactions governing microbial population dynamics.

Data accessibility. A pair of files (Mathematica notebook for structural identifiability of two- to five-species generalized Lotka–Volterra models with relative abundance data and R code for the application to synthetic community data) with code has been included as electronic supplementary material.

Authors' contributions. C.H.R. and B.J.R. conceived the study; C.H.R., M.J.E. and B.J.R. performed analyses and wrote the paper.

Competing interests. We declare we have no competing interests.

Funding. Support for this project comes from National Institutes of Health grant no. P20GM104420.

Acknowledgements. We thank the CMCI Microbiome working group for useful discussion.

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
