## [Peer Review File · Royal Society Open Science]

Review History

RSOS-191460.R0 (Original submission)

Review form: Reviewer 1

Is the manuscript scientifically sound in its present form?

No

Are the interpretations and conclusions justified by the results?

No

Is the language acceptable?

Yes

Do you have any ethical concerns with this paper?

No

Have you any concerns about statistical analyses in this paper?

No

Recommendation?

Reject

Comments to the Author(s)

The authors present an analysis on identifiability of model parameters within the context of microbial population dynamics. Therefore, they use a so-called generalized Lotka-Volterra system, an ordinary differential equation model with two or more species and so-called relative abundance data, i.e. time courses of relative abundances of the model species. Within their study, they try to draw conclusions on the identifiability, i.e. if it is possible 'in principle' to estimate the parameters from relative data, instead of absolute abundances or numbers of individuals of the species. Within their analysis, they reformulate their system under the constraint of relative data and make statements about the general structural identifiability of certain parameter classes within the generalized Lotka-Volterra system. The further show numerical simulations of a small example model with three different parameter scenarios, which should support and verify their findings of generally identifiable parameter classes.

Major points:

* Although, the central point of the presented work should be the structural identifiability, its definition within the manuscript remains mathematically vague and is easily overlooked when reading the 'Analysis' section. There is also a lack of precise definition and discussion on the discrimination between structural and practical identifiability. Perhaps a proper 'Methods' Section with additional information on the applied methods, techniques and definitions would be beneficial.

* l. 217-218: In some definitions of practical non-identifiability, it explicitly does not depend on the system, but only on the quality and amount of available data. I suggest the authors, to distinguish between practical and structural (non-)identifiability in the beginning of the article and to clarify right usage of the terms in lines 217ff

* The introduction is quite lengthy compared to the information contained and exhibits numerous repetitions, whereas some terms and circumstances are not well explained. There is a lot of repeated information, what is to be covered in the 'following' or 'in this section', concerning the structural identifiability and the relative abundances. The same applies for the 'Implications of Structural Identifiability Analysis' Section. On the other hand, results are presented and discussed only briefly. For example, the figures with the numerical results are not well described in the text so that they can support the authors' findings. Furthermore, there is fair amount of concepts, expressions, methods, technicalities which are not well explained in the main text or suffer from references, making it hard for readers from a different or neighboring field to follow and judge the appropriateness of the argumentation, e.g.: process error, carrying capacity and negative transition rate parameters $\beta_{i,j}$, likelihood surface / (maximum) likelihood, communities / community data, amplicon read out, MVN, pomp function

*l. 98: The authors state that there exist several methods. Is the presented methodology an application or extension of an existing method or a new concept?

*The central 'mathematical trick' in lines 120 - 125 seems to be rather trivial on the one hand, but its applicability and implications are not well enough explained although it guarantees the key finding of the study. (i.e. why are the coefficients of the DAE identifiable?)

* The rigorousness for the extrapolation for models with 3, 4 and 5 species is weak and the universality of the statement in l. 127-128 for any larger model is not proven.

MINOR

l. 27 How can the change over time in populations be referred to the prevention of illnesses?

* In my opinion, the term 'constant', e.g. in l. 125 is miss-leading, as it could be confused with an additive constant. Indeed, it rather is a factor, scaling, scaling factor, or a scaling constant.

* l 153-164: The processes from which the two error types are generated seem to be quite specific and it remains unclear if this level of detail is required for the drawn conclusions about the structural identifiability is necessary. If this is a standard choice within the field, it should be stated and/or explained why these processes are chosen. How much does the result depend on these specific processes and would a e.g. gaussian distributed error yield comparable results?

* 167. The third scenario needs more explanation

* l. 171: Is indeed the 'verification' of the convergence the desired task here or is it rather intended to 'ensure' the convergence of the fit by initializing it close to the 'true' value?

* l. 165 - 185: I like on the one hand that a lot of technical details on the fitting procedure are given. On the other hand, it is hard to understand if this influences the results. The technical details should be rather part of a proper 'Methods' section.

* l. 186f: It is not well explained how this conclusion can be drawn from the Figures

* l. 204-214: I don't see why this technique should be discussed here, as it cannot be applied on the presented model class / specification.

* Equation 2: primed variables are not defined as derivatives with respect to time.

* Equation 2: Why is the derivative of the total number stated here? Should it maybe rather be just the total number $N = X + Y$? Is the change of the total number over time not constant?

* Equation 6: not aligned and thus hard to read

* typo l. 107: analyses uncovers -> analyses uncover

* Figure 1: Some colors of the arrows are too light. Is this parameter set 'Value 1' or 'Value 2'?

* Figure 2: It is hard to understand which graph corresponds to which values and where this supports the results.

* Figure 3: The green line is barely visible. The Relation to the main text unclear. There is no Methods section as stated in the caption.

Review form: Reviewer 2

Is the manuscript scientifically sound in its present form?

Yes

Are the interpretations and conclusions justified by the results?

No

Is the language acceptable?

Yes

Do you have any ethical concerns with this paper?

No

Have you any concerns about statistical analyses in this paper?

No

Recommendation?

Major revision is needed (please make suggestions in comments)

Comments to the Author(s)

The authors investigate parameter identifiability of the G-LV model for microbiome studies using structural identifiability analysis and model fitting to synthetic data, in both situations where absolute abundance data and relative abundance data are available.

In recent years, several models and methods have been proposed to detect and estimate OTUs interactions from microbiota sequencing data. It is frequent that available data only relates to relative abundances. Given the dimension of the problem, the question of the identifiability of interaction parameters is of major importance. This is, to my knowledge, the first study attempting to investigate this problem. The results suggest that it is actually possible to estimate so, despite only relative interactions may be estimated when working with relative abundances. This is an exciting and very useful study, however, a few components are missing. The results in their present form do not fully support the conclusions.

Major comments

- Equation (0) line 73. The model equation contains a term related to exogenous variables which is never used in this study, nor discussed. For that reason, I think this term should be removed.
- The model as detailed on lines 71-82 assumes a species-specific carrying capacity K_i forced by a term of within-species interaction β_{ii} . Another option chosen in some studies is to assume a global carrying capacity K , which may be more realistic (eg in Gao et al frontiers in Microbiology 2018). What is the consequence of one choice or the other on parameters estimation? This should be discussed.
- Equation (6) is not very informative for the reader and should be moved to supplementary. Would the input output DAE in terms of x , x' and x'' whose coefficients are identifiable be more useful for the reader here?
- Strong assumptions were made in the estimation procedure: (1) regarding starting points, chosen around the true values; (2) regarding initial populations sizes. I may have missed something but my understanding is that the full space of parameters may therefore not be explored, impeding the possibility to find other maximums of the likelihood surface. Why starting there and not exploring the full space? What if the population size is not initialize at a realistic value?
- Fig1. Usually, the arrows looping on nodes should include both the β_{ii} term and the growth rate; which, I think, is not the case here
- Choice of figures could be clearly improved. For example, fig 2 contains 4 different graphs but globally very little information, the aim being, if I understand correctly, to illustrate that different population sizes can lead to similar relative abundances. This is kind of obvious but if the authors wish to illustrate it, I think 4 subgraphs are not necessary.
- In Figure 3, it looks like the stochasticity present in the simulations is very low. Despite this is the case for both processes, the process error seems very low. How was the simulated value chosen? How realistic is it compared with the variability found in real OTU data?

- If I understand well, the aim of Figure 4 is to illustrate the possibility of estimating parameters from different kind of abundances. It is therefore a key figure of the paper. Is the fit shown actually good? Is there a way to quantify the fit quality? What about the estimation errors? Is the figure actually demonstrating that estimates are correct on the relative scale? It is really not obvious and should be improved/clarified.

- In the paper, only two different sets of parameters are assessed. Such a study would be really useful if a more global study was carried out, simulating data on a range of parameters set over realistic parameters intervals, and estimating associated parameters. In particular, the sensitivity and specificity at detecting significant interactions (in the correct direction) would be an interesting measure of efficacy.

- How does stochasticity in the simulation model affect estimates? In microbiota studies, usually a limited number of points is available (5, 10, 20?). How does this number affect the reliability of the estimates? What is the minimum number of points required? This should be discussed.

Minor comments

- Abstract: the abstract could be improved. In particular, the sentence "relative abundance data alone do contain information on relative interaction strengths..." is nearly repeated 6 lines below.

- First equation is not numbered

- Line 107, analyses (...) uncover

- Figure 4: tracing the 0 line would be useful for the reader.

Decision letter (RSOS-191460.R0)

31-Jan-2020

Dear Dr Remien:

Manuscript ID RSOS-191460 entitled "Parameter Identifiability of the Generalized Lotka-Volterra Model for Microbiome Studies" which you submitted to Royal Society Open Science, has been reviewed. The comments from reviewers are included at the bottom of this letter.

In view of the criticisms of the reviewers, the manuscript has been rejected in its current form. However, a new manuscript may be submitted which takes into consideration these comments.

Please note that resubmitting your manuscript does not guarantee eventual acceptance, and that your resubmission will be subject to peer review before a decision is made.

Your resubmitted manuscript should be submitted by 30-Jul-2020. If you are unable to submit by this date please contact the Editorial Office.

on behalf of Dr Dirk Drasdo (Associate Editor) and Mark Chaplain (Subject Editor)
openscience@royalsociety.org

Associate Editor Comments to Author (Dr Dirk Drasdo):

Associate Editor: 1

Comments to the Author:

We have now received two referee reports for your submission. Please ensure that you read and address all comments within these reports thoroughly, and provide a response each comment in a point-by-point response letter upon resubmission.

Specifically, both Referee #1 and #2 state that several aspects of your manuscript are not well described, and need further explanation, discussion, and clarification. Please ensure that you address the referees' comments throughout your manuscript.

Furthermore, Referee #2 had several comments regarding the presentation of the figures within your manuscript, and stated that the 'choice of figures could be clearly improved'. Please ensure that you address these and other concerns.

Reviewers' Comments to Author:

Reviewer: 1

Comments to the Author(s)

The authors present an analysis on identifiability of model parameters within the context of microbial population dynamics. Therefore, they use a so-called generalized Lotka-Volterra system, an ordinary differential equation model with two or more species and so-called relative abundance data, i.e. time courses of relative abundances of the model species. Within their study, they try to draw conclusions on the identifiability, i.e. if it is possible 'in principle' to estimate the parameters from relative data, instead of absolute abundances or numbers of individuals of the species. Within their analysis, they reformulate their system under the constraint of relative data and make statements about the general structural identifiability of certain parameter classes within the generalized Lotka-Volterra system. They further show numerical simulations of a small example model with three different parameter scenarios, which should support and verify their findings of generally identifiable parameter classes.

Major points:

* Although, the central point of the presented work should be the structural identifiability, its definition within the manuscript remains mathematically vague and is easily overlooked when reading the 'Analysis' section. There is also a lack of precise definition and discussion on the discrimination between structural and practical identifiability. Perhaps a proper 'Methods' Section with additional information on the applied methods, techniques and definitions would be beneficial.

* 1. 217-218: In some definitions of practical non-identifiability, it explicitly does not depend on the system, but only on the quality and amount of available data. I suggest the authors, to

distinguish between practical and structural (non-)identifiability in the beginning of the article and to clarify right usage of the terms in lines 217ff

* The introduction is quite lengthy compared to the information contained and exhibits numerous repetitions, whereas some terms and circumstances are not well explained. There is a lot of repeated information, what is to be covered in the 'following' or 'in this section', concerning the structural identifiability and the relative abundances. The same applies for the 'Implications of Structural Identifiability Analysis' Section. On the other hand, results are presented and discussed only briefly. For example, the figures with the numerical results are not well described in the text so that they can support the authors' findings. Furthermore, there is fair amount of concepts, expressions, methods, technicalities which are not well explained in the main text or suffer from references, making it hard for readers from a different or neighboring field to follow and judge the appropriateness of the argumentation, e.g.: process error, carrying capacity and negative transition rate parameters $\beta_{i,j}$, likelihood surface / (maximum) likelihood, communities / community data, amplicon read out, MVN, pomp function

* l. 98: The authors state that there exist several methods. Is the presented methodology an application or extension of an existing method or a new concept?

* The central 'mathematical trick' in lines 120 - 125 seems to be rather trivial on the one hand, but its applicability and implications are not well enough explained although it guarantees the key finding of the study. (i.e. why are the coefficients of the DAE identifiable?)

* The rigorousness for the extrapolation for models with 3, 4 and 5 species is weak and the universality of the statement in l. 127-128 for any larger model is not proven.

MINOR

l. 27 How can the change over time in populations be referred to the prevention of illnesses?

* In my opinion, the term 'constant', e.g. in l. 125 is miss-leading, as it could be confused with an additive constant. Indeed, it rather is a factor, scaling, scaling factor, or a scaling constant.

* l 153-164: The processes from which the two error types are generated seem to be quite specific and it remains unclear if this level of detail is required for the drawn conclusions about the structural identifiability is necessary. If this is a standard choice within the field, it should be stated and/or explained why these processes are chosen. How much does the result depend on these specific processes and would a e.g. gaussian distributed error yield comparable results?

* 167. The third scenario needs more explanation

* l. 171: Is indeed the 'verification' of the convergence the desired task here or is it rather intended to 'ensure' the convergence of the fit by initializing it close to the 'true' value?

* l. 165 - 185: I like on the one hand that a lot of technical details on the fitting procedure are given. On the other hand, it is hard to understand if this influences the results. The technical details should be rather part of a proper 'Methods' section.

* l. 186f: It is not well explained how this conclusion can be drawn from the Figures

* l. 204-214: I don't see why this technique should be discussed here, as it cannot be applied on the presented model class / specification.

- * Equation 2: primed variables are not defined as derivatives with respect to time.
- * Equation 2: Why is the derivative of the total number stated here? Should it maybe rather be just the total number $N = X + Y$? Is the change of the total number over time not constant?
- * Equation 6: not aligned and thus hard to read
- * typo l. 107: analyses uncovers -> analyses uncover
- * Figure 1: Some colors of the arrows are too light. Is this parameter set 'Value 1' or 'Value 2'?
- * Figure 2: It is hard to understand which graph corresponds to which values and where this supports the results.
- * Figure 3: The green line is barely visible. The Relation to the main text unclear. There is no Methods section as stated in the caption.

Reviewer: 2

Comments to the Author(s)

The authors investigate parameter identifiability of the G-LV model for microbiome studies using structural identifiability analysis and model fitting to synthetic data, in both situations where absolute abundance data and relative abundance data are available.

In recent years, several models and methods have been proposed to detect and estimate OTUs interactions from microbiota sequencing data. It is frequent that available data only relates to relative abundances. Given the dimension of the problem, the question of the identifiability of interaction parameters is of major importance. This is, to my knowledge, the first study attempting to investigate this problem. The results suggest that it is actually possible to estimate so, despite only relative interactions may be estimated when working with relative abundances. This is an exciting and very useful study, however, a few components are missing. The results in their present form do not fully support the conclusions.

Major comments

- Equation (0) line 73. The model equation contains a term related to exogenous variables which is never used in this study, nor discussed. For that reason, I think this term should be removed.
- The model as detailed on lines 71-82 assumes a species-specific carrying capacity K_i forced by a term of within-species interaction β_{ii} . Another option chosen in some studies is to assume a global carrying capacity K , which may be more realistic (eg in Gao et al *frontiers in Microbiology* 2018). What is the consequence of one choice or the other on parameters estimation? This should be discussed.
- Equation (6) is not very informative for the reader and should be moved to supplementary. Would the input output DAE in terms of x , x' and x'' whose coefficients are identifiable be more useful for the reader here?
- Strong assumptions were made in the estimation procedure: (1) regarding starting points, chosen around the true values; (2) regarding initial populations sizes. I may have missed something but my understanding is that the full space of parameters may therefore not be explored, impeding the possibility to find other maximums of the likelihood surface. Why starting there and not exploring the full space? What if the population size is not initialize at a realistic value?
- Fig1. Usually, the arrows looping on nodes should include both the β_{ii} term and the growth rate; which, I think, is not the case here

- Choice of figures could be clearly improved. For example, fig 2 contains 4 different graphs but globally very little information, the aim being, if I understand correctly, to illustrate that different population sizes can lead to similar relative abundances. This is kind of obvious but if the authors wish to illustrate it, I think 4 subgraphs are not necessary.
- In Figure 3, it looks like the stochasticity present in the simulations is very low. Despite this is the case for both processes, the process error seems very low. How was the simulated value chosen? How realistic is it compared with the variability found in real OTU data?
- If I understand well, the aim of Figure 4 is to illustrate the possibility of estimating parameters from different kind of abundances. It is therefore a key figure of the paper. Is the fit shown actually good? Is there a way to quantify the fit quality? What about the estimation errors? Is the figure actually demonstrating that estimates are correct on the relative scale? It is really not obvious and should be improved/clarified.
- In the paper, only two different sets of parameters are assessed. Such a study would be really useful if a more global study was carried out, simulating data on a range of parameters set over realistic parameters intervals, and estimating associated parameters. In particular, the sensitivity and specificity at detecting significant interactions (in the correct direction) would be an interesting measure of efficacy.
- How does stochasticity in the simulation model affect estimates? In microbiota studies, usually a limited number of points is available (5, 10, 20?). How does this number affect the reliability of the estimates? What is the minimum number of points required? This should be discussed.

Minor comments

- Abstract: the abstract could be improved. In particular, the sentence "relative abundance data alone do contain information on relative interaction strengths..." is nearly repeated 6 lines below.
- First equation is not numbered
- Line 107, analyses (...) uncover
- Figure 4: tracing the 0 line would be useful for the reader.

Author's Response to Decision Letter for (RSOS-191460.R0)

See Appendix A.

RSOS-201378.R0

Review form: Reviewer 1

Is the manuscript scientifically sound in its present form?

Yes

Are the interpretations and conclusions justified by the results?

Yes

Is the language acceptable?

Yes

Do you have any ethical concerns with this paper?

No

Have you any concerns about statistical analyses in this paper?

No

Recommendation?

Major revision is needed (please make suggestions in comments)

Comments to the Author(s)

Compared to the first version of the manuscript, the authors reworked all mentioned points so that readability improved significantly. At least from my point of view, it much easier to follow the author's intentions. Further, they now clearly state that the first part, i.e. the structural identifiability analysis, is an application of an existing method that they apply to a new field and model. The conclusion drawn from this section is that model parameters can be estimated from relative data. As this is not the standard approach for microbiome data and gLV models, this is a noteworthy finding for this field.

However, the section on the analysis of 'local practical identifiability' is rather an example of such a gLV model with relative data that a broad analysis of local practical identifiability. Here, the authors generate data of one noise realisation for simulated measurements and apply a particle MCMC search to fit the model to the simulated data. The fit to the "Value 2" scenario shows obvious deviations from the trajectory of the underlying truth. Further, they show boxplots of the parameter values from the best MCMC runs to conclude that the likelihood has a unique maximum in the local neighborhood. In my eyes, only vague conclusions about the (local) practical identifiability can be drawn from this boxplot analysis. The box plots from the 'value 2' scenario are quite broad and rather support a non-unique maximum of the likelihood, i.e. a practical non-identifiability. At least histograms of the parameter values and/or their actual corresponding likelihood values are needed to draw conclusions about the mono-modality of the distributions and thus of the uniqueness of the likelihood's maximum. I do not doubt that these models and parameters may be (locally) practical identifiable, but in my eyes the presented analyzes are not strong enough to support this.

Minor points:

- l. 99 and other appearances: only 'Saccomani' is the last name of the author in the referenced work, Pia might be the middle name

- l. 129ff : There is something like a sentence missing between equation (6) and equations (7). Where does equation (7) follow from?

- equations (7): $x(t)$ should be simply x

- l. 133/134 I still find it difficult to follow the authors how they extract the coefficients from these lines from equation (7)

- l. 142 and 149: same sentence used

- l. 151 and 164: convergence of what?

- l. 167: There is no parameter N_0 in Table 1. Also, it remains unclear if parameter value set 1 or 2 is used as reference for scenario 3. Would it be possible to simply expand table 1 by a column with the parameter set used for scenario 3?

- 1.184 only the stated simultaneous combination of both error sources is analyzed in this work, not their single contributions.

- 1. 212-223 could be shortened

Review form: Reviewer 2

Is the manuscript scientifically sound in its present form?

Yes

Are the interpretations and conclusions justified by the results?

Yes

Is the language acceptable?

Yes

Do you have any ethical concerns with this paper?

No

Have you any concerns about statistical analyses in this paper?

No

Recommendation?

Accept with minor revision (please list in comments)

Comments to the Author(s)

I thank the authors for their point by point answers and updated version of the manuscript which I think is really improved. Figures changes are also notable and definitely improve clarity. Again, I believe this is an important topic and very useful work. However, I would like to stress a remaining aspect for which I did not find a proper answer in the revised version and strongly believe this is a key point for the scientific community.

Following the comment regarding a larger analysis over the parameter space, the authors replied that they only aimed here at investigating likelihood estimation locally around the true known parameters. Despite I understand that noisy data can generate a lack of identifiability within a small interval around the true value, and that investigating such bias is important; of more importance is, I think, the fact that noisy data and availability of relative counts may lead to select completely different sets of parameters combinations. When such a combination of interactions generates a good likelihood, it may lead to another type of bias, of I think much bigger impact. This latter is, to my mind, a very important point that should not be eluded. This is in particular a key question for those who analyze in practice such systems and data. It is not clear to me why such exploration is not done as the package used here, POMP, could enable to analyze likelihood in wider spaces similarly.

If this mentioned global analysis of the parameter space is not included in the present paper, I suggest the title to be modified to indicate that the article focuses on the question of bias and identifiability locally.

Decision letter (RSOS-201378.R0)

Dear Dr Remien

The Editors assigned to your paper RSOS-201378 "Parameter Identifiability of the Generalized Lotka-Volterra Model for Microbiome Studies" have now received comments from reviewers and would like you to revise the paper in accordance with the reviewer comments and any comments from the Editors. Please note this decision does not guarantee eventual acceptance.

Please submit your revised manuscript and required files (see below) no later than 21 days from today's (ie 18-Jan-2021) date. Note: the ScholarOne system will 'lock' if submission of the revision is attempted 21 or more days after the deadline. If you do not think you will be able to meet this deadline please contact the editorial office immediately.

on behalf of Dr Dirk Drasdo (Associate Editor) and Mark Chaplain (Subject Editor)
openscience@royalsociety.org

Reviewer comments to Author:

Reviewer: 1

Comments to the Author(s)

Compared to the first version of the manuscript, the authors reworked all mentioned points so that readability improved significantly. At least from my point of view, it much easier to follow the author's intentions. Further, they now clearly state that the first part, i.e. the structural

identifiability analysis, is an application of an existing method that they apply to a new field and model. The conclusion drawn from this section is that model parameters can be estimated from relative data. As this is not the standard approach for microbiome data and gLV models, this is a noteworthy finding for this field.

However, the section on the analysis of 'local practical identifiability' is rather an example of such a gLV model with relative data that a broad analysis of local practical identifiability. Here, the authors generate data of one noise realisation for simulated measurements and apply a particle MCMC search to fit the model to the simulated data. The fit to the "Value 2" scenario shows obvious deviations from the trajectory of the underlying truth. Further, they show boxplots of the parameter values from the best MCMC runs to conclude that the likelihood has a unique maximum in the local neighborhood. In my eyes, only vague conclusions about the (local) practical identifiability can be drawn from this boxplot analysis. The box plots from the 'value 2' scenario are quite broad and rather support a non-unique maximum of the likelihood, i.e. a practical non-identifiability. At least histograms of the parameter values and/or their actual corresponding likelihood values are needed to draw conclusions about the mono-modality of the distributions and thus of the uniqueness of the likelihood's maximum. I do not doubt that these models and parameters may be (locally) practical identifiable, but in my eyes the presented analyzes are not strong enough to support this.

Minor points:

- l.99 and other appearances: only 'Saccomani' is the last name of the author in the referenced work, Pia might be the middle name
- l. 129ff : There is something like a sentence missing between equation (6) and equations (7). Where does equation (7) follow from?
- equations (7): $x(t)$ should be simply x
- l. 133/134 I still find it difficult to follow the authors how they extract the coefficients from these lines from equation (7)
- l. 142 and 149: same sentence used
- l. 151 and 164: convergence of what?
- l. 167: There is no parameter N_0 in Table 1. Also, it remains unclear if parameter value set 1 or 2 is used as reference for scenario 3. Would it be possible to simply expand table 1 by a column with the parameter set used for scenario 3?
- l.184 only the stated simultaneous combination of both error sources is analyzed in this work, not their single contributions.
- l. 212-223 could be shortened

Reviewer: 2

Comments to the Author(s)

I thank the authors for their point by point answers and updated version of the manuscript which I think is really improved. Figure changes are also notable and definitely improve clarity. Again, I believe this is an important topic and very useful work. However, I would like to stress a remaining aspect for which I did not find a proper answer in the revised version and strongly believe this is a key point for the scientific community.

Following the comment regarding a larger analysis over the parameter space, the authors replied that they only aimed here at investigating likelihood estimation locally around the true known parameters. Despite I understand that noisy data can generate a lack of identifiability within a small interval around the true value, and that investigating such bias is important; of more importance is, I think, the fact that noisy data and availability of relative counts may lead to select completely different sets of parameters combinations. When such a combination of interactions generates a good likelihood, it may lead to another type of bias, of I think much bigger impact. This latter is, to my mind, a very important point that should not be eluded. This is in particular a key question for those who analyze in practice such systems and data. It is not clear to me why such exploration is not done as the package used here, POMP, could enable to analyze likelihood in wider spaces similarly.

If this mentioned global analysis of the parameter space is not included in the present paper, I suggest the title to be modified to indicate that the article focuses on the question of bias and identifiability locally.

===PREPARING YOUR MANUSCRIPT===

===PREPARING YOUR REVISION IN SCHOLARONE===

Author's Response to Decision Letter for (RSOS-201378.R0)

See Appendix B.

RSOS-201378.R1 (Revision)

Review form: Reviewer 1

Is the manuscript scientifically sound in its present form?

Yes

Are the interpretations and conclusions justified by the results?

Yes

Is the language acceptable?

Yes

Do you have any ethical concerns with this paper?

No

Have you any concerns about statistical analyses in this paper?

No

Recommendation?

Accept as is

Comments to the Author(s)

I would like to thank the authors for their detailed response to the updated manuscript. From my point of view, the changes made to the manuscript now clearly allow to separate the structural identifiability analysis from the analysis of practical identifiability in the presented microbiome model. In addition, the altered title of the manuscript now allows for a better classification of the core of the presented work. Further, the extended discussion about practical issues of parameter estimation and identifiability analysis using noisy and potentially sparse data adequately addresses the illustrative character of the 'local practical identifiability' section.

In conclusion, I would rate the presented work as an interesting contribution to the field and I would consider the updated version of the manuscript to be ready for acceptance.

Decision letter (RSOS-201378.R1)

Dear Dr Remien,

It is a pleasure to accept your manuscript entitled "Structural Identifiability of the Generalized Lotka-Volterra Model for Microbiome Studies" in its current form for publication in Royal Society Open Science. The comments of the reviewer(s) who reviewed your manuscript are included at the foot of this letter.

You can expect to receive a proof of your article in the near future. Please contact the editorial office (openscience@royalsociety.org) and the production office

(openscience_proofs@royalsociety.org) to let us know if you are likely to be away from e-mail contact – if you are going to be away, please nominate a co-author (if available) to manage the proofing process, and ensure they are copied into your email to the journal.

on behalf of Dr Dirk Drasdo (Associate Editor) and Mark Chaplain (Subject Editor)
openscience@royalsociety.org

Reviewer comments to Author:
Reviewer: 1

Comments to the Author(s)

I would like to thank the authors for their detailed response to the updated manuscript. From my point of view, the changes made to the manuscript now clearly allow to separate the structural identifiability analysis from the analysis of practical identifiability in the presented microbiome model. In addition, the altered title of the manuscript now allows for a better classification of the core of the presented work. Further, the extended discussion about practical issues of parameter estimation and identifiability analysis using noisy and potentially sparse data adequately addresses the illustrative character of the 'local practical identifiability' section. In conclusion, I would rate the presented work as an interesting contribution to the field and I would consider the updated version of the manuscript to be ready for acceptance.

Appendix A

July 29, 2020

Dear Drs. Drasdo and Chaplain,

Thank you for the encouraging response to our manuscript, "Parameter Identifiability of the Generalized Lotka-Volterra Model for Microbiome Studies." Both reviewers felt that several aspects of the manuscript needed further explanation and clarification. In particular, we modified the text throughout the manuscript for clarity. Specifically, we added a paragraph to the introduction describing parameter identifiability analysis, added references and description for the structural identifiability methods, and completely reworked the local practical identifiability section (this section seemed to draw the most confusion from the Reviewers). We also modified the figures as suggested by the second reviewer. Specifically, we deleted figures 1 and 3 from our original submission, changed color schemes for clarity, and added titles and descriptions which should aid in the interpretation of the remaining figures. In addition to these larger changes, we addressed the other more minor comments made by the reviewers. Details of these changes are as follows:

Reviewer 1

Although, the central point of the presented work should be the structural identifiability, its definition within the manuscript remains mathematically vague and is easily overlooked when reading the 'Analysis' section. There is also a lack of precise definition and discussion on the discrimination between structural and practical identifiability.

We agree and added a paragraph introducing the concepts of structural and practical identifiability in the Introduction section lines 57-68. We have also included references here that review these topics.

I. 217-218: In some definitions of practical non-identifiability, it explicitly does not depend on the system, but only on the quality and amount of available data. I suggest the authors, to distinguish between practical and structural (non-)identifiability in the beginning of the article and to clarify right usage of the terms in lines 217ff

We clarified the terminology in the new paragraph of the introduction, lines 57-68.

The introduction is quite lengthy compared to the information contained and exhibits numerous repetitions, whereas some terms and circumstances are not well explained.

There is a lot of repeated information, what is to be covered in the 'following' or 'in this section', concerning the structural identifiability and the relative abundances.

We eliminated the redundancies and repeated information as suggested.

The same applies for the 'Implications of Structural Identifiability Analysis' Section. On the other hand, results are presented and discussed only briefly. For example, the figures with the numerical results are not well described in the text so that they can support the authors' findings. Furthermore, there is fair amount of concepts, expressions, methods, technicalities which are not well explained in the main text or suffer from references, making it hard for readers from a different or neighboring field to follow and judge the appropriateness of the argumentation, e.g.: process error, carrying capacity and negative

transition rate parameters $\beta_{i,j}$, likelihood surface / (maximum) likelihood, communities / community data, amplicon read out, MVN, pomp function

Again, we tried to eliminate any repetition. We also added descriptions to the methods and concepts, specifically the references to the structural identifiability methods in lines 109-110 and the suggested POMP protocol on lines 170-171.

The authors state that there exist several methods. Is the presented methodology an application or extension of an existing method or a new concept?

The presented methodology is the application of an existing method to a new model. This is now clarified in lines 108-110.

The central 'mathematical trick' in lines 120 - 125 seems to be rather trivial on the one hand, but its applicability and implications are not well enough explained although it guarantees the key finding of the study. (i.e. why are the coefficients of the DAE identifiable?)

For an overview see Audoly et al 2001 (which we now refer the reader to on lines 109-110). The idea is to write an input-output equation (a monic differential polynomial only in terms of measured variables and unknown parameters). The coefficients of input-output equations are identifiable combinations of parameters (Audoly et al., 2001).

The rigorousness for the extrapolation for models with 3, 4 and 5 species is weak and the universality of the statement in l. 127-128 for any larger model is not proven.

We agree that the extrapolation to models with an arbitrary number of species is a conjecture and have explicitly said this now on line 140. While we agree that this is not rigorously proven, we feel that the evidence for the pattern is strong.

How can the change over time in populations be referred to the prevention of illnesses?

There are numerous ways that a microbial species changing in abundance over time can lead to illness. A direct example is that of *Clostridium difficile* (aka C-diff), which causes severe damage to the colon. If species that prevent *C. difficile* from increasing in abundance are identified, they may be useful for prevention of illness.

In my opinion, the term 'constant', e.g. in l. 125 is miss-leading, as it could be confused with an additive constant. Indeed, it rather is a factor, scaling, scaling factor, or a scaling constant.

Agreed; we modified this language.

l 153-164: The processes from which the two error types are generated seem to be quite specific and it remains unclear if this level of detail is required for the drawn conclusions about the structural identifiability is necessary. If this is a standard choice within the field, it should be stated and/or explained why these processes are chosen. How much does the result depend on these specific processes and would a e.g. gaussian distributed error yield comparable results?

Indeed, there are numerous choices that could be made to impose noise into the system, and the results will naturally depend on the level of noise. We attempted to create measurement noise in a way that is consistent with the generating process (DNA sequencing). We modified the language throughout this section to clarify our intentions. Here, we simply wanted to verify that the addition of small amounts of noise does not necessarily fundamentally alter the structural identifiability results (which are the main results of the paper).

167. The third scenario needs more explanation

We added some more explanation on lines 167-170. Briefly, the idea is to show that the population size relative to the initial population size can actually be estimated from relative abundance data (at least in this example).

l. 171: Is indeed the 'verification' of the convergence the desired task here or is it rather intended to 'ensure' the convergence of the fit by initializing it close to the 'true' value?

Verification is correct. We changed the language throughout this section to clarify that we are performing a *local* practical identifiability analysis near the true parameter values. In essence, we wish to verify that there is indeed a maximum in the likelihood surface near true parameter values.

l. 165 - 185: I like on the one hand that a lot of technical details on the fitting procedure are given. On the other hand, it is hard to understand if this influences the results. The technical details should be rather part of a proper 'Methods' section.

We are happy to move this and other details to a methods section if the editor feels it would improve clarity. We clarified in the text on line 170 that these details are following the suggested protocol for POMP.

l. 186f: It is not well explained how this conclusion can be drawn from the Figures

We added a sentence to clarify this on lines 185-186. Briefly, the boxplots show that there is indeed a local maximum in the likelihood surface near the true parameter values.

** l. 204-214: I don't see why this technique should be discussed here, as it cannot be applied on the presented model class / specification.*

We feel that it is essential to compare our results to the standard fitting routines that use linear regression (and assume only process error). Linear regression is scalable to estimate parameters in large microbial communities with large number of unknown parameters. In contrast, the relative abundance equations do not have an analogous linear form so the fitting algorithm will necessarily be more complex. We do not claim to have solved this problem in this paper but want to point out the issue and compare results to the current state of the art.

** Equation 2: primed variables are not defined as derivatives with respect to time.*

Fixed.

** Equation 2: Why is the derivative of the total number stated here? Should it maybe rather be just the total number $N = X + Y$? Is the change of the total number over time not constant?*

The total population size is not constant over time. The derivative of N is included because it is used in equations 5 to obtain equation 6.

** Equation 6: not aligned and thus hard to read*

We agree that equation 6 is cumbersome (and this is just the 2 species case). It is currently not aligned because we attempted to use alignment to link terms within parentheses. We hope that final typesetting by the journal can aid in clarity; we are also willing moving this to an Appendix if deemed appropriate.

** typo l. 107: analyses uncovers -> analyses uncover*

Fixed.

** Figure 1: Some colors of the arrows are too light. Is this parameter set 'Value 1' or 'Value 2'?*

Fixed and clarified the parameter sets.

** Figure 2: It is hard to understand which graph corresponds to which values and where this supports the results.*

Fixed.

** Figure 3: The green line is barely visible. The Relation to the main text unclear. There is no Methods section as stated in the caption.*

Fixed. The methods section references was a relic from a previous version.

Reviewer: 2

Major comments

- Equation (0) line 73. The model equation contains a term related to exogenous variables which is never used in this study, nor discussed. For that reason, I think this term should be removed.

We agree and removed the term.

*- The model as detailed on lines 71-82 assumes a species-specific carrying capacity K_i forced by a term of within-species interaction β_{ii} . Another option chosen in some studies is to assume a global carrying capacity K , which may be more realistic (eg in Gao et al *frontiers in Microbiology* 2018). What is the consequence of one choice or the other on parameters estimation? This should be discussed.*

We did a structural identifiability analysis for a small system with a global carrying capacity and found a similar result---not all parameters are identifiable. The interactions parameters and carrying capacity

are identifiable up to a scaling factor. If the carrying capacity is known (or assumed to be 1), then the model is identifiable. This is analogous to the results presented in that if a single interaction is known (or equivalently the carrying capacity of a single species), then the model is identifiable. We feel that choosing a global carrying capacity K is a bit extraneous since what we presented is more general (a global carrying capacity is a special case of our results) and the results do not fundamentally change with this parameterization. However, we are happy to include these results if the editor feels the paper would benefit from their inclusion.

Other

- Equation (6) is not very informative for the reader and should be moved to supplementary. Would the input output DAE in terms of x , x' and x'' whose coefficients are identifiable be more useful for the reader here?

We completely agree that equation 6 is cumbersome. Unfortunately, the input-output DAE is equally cumbersome (or perhaps even worse). As mentioned before, we are happy to move this to a supplementary if the editor feels it would benefit the paper.

- Strong assumptions were made in the estimation procedure: (1) regarding starting points, chosen around the true values; (2) regarding initial populations sizes. I may have missed something but my understanding is that the full space of parameters may therefore not be explored, impeding the possibility to find other maximums of the likelihood surface. Why starting there and not exploring the full space?

We reworked this section for clarity (lines 141-192). Briefly, we are performing a *local* (rather than global) practical parameter identifiability analysis near the true parameters. The goal is to identify whether there is a maximum located near the true parameters (rather than, say, a ridge). Even for a small system like we have presented, there are a large number of parameters making exploration of the full parameter space cumbersome.

What if the population size is not initialize at a realistic value?

If the population size is not initialized at a realistic value, then interaction parameters are estimated up to a scaling factor. This point is clarified in the text in the "third scenario" line 167.

- Fig1. Usually, the arrows looping on nodes should include both the β_{ii} term and the growth rate; which, I think, is not the case here

We decided that figure 1 did not add a lot of value and was removed (though we can add it back in with such loops if the editor feels it would improve the manuscript).

- Choice of figures could be clearly improved. For example, fig 2 contains 4 different graphs but globally very little information, the aim being, if I understand correctly, to illustrate that different population sizes can lead to similar relative abundances. This is kind of obvious but if the authors wish to illustrate it, I think 4 subgraphs are not necessary.

We modified the previous figure 2 for clarity. We agree that it is obvious that different population sizes can lead to similar relative abundances, but it is not immediately obvious how parameters have to scale

with different population sizes to lead to identical relative abundances. This result follows from the structural identifiability analysis.

- In Figure 3, it looks like the stochasticity present in the simulations is very low. Despite this is the case for both processes, the process error seems very low. How was the simulated value chosen? How realistic is it compared with the variability found in real OTU data?

The amount of process and measurement noise included (and indeed the parameter values for the synthetic community) are somewhat arbitrary. We clarified our intentions in the “Local Practical Identifiability in a Synthetic Community” section, which is entirely reworked. It is unclear the degree to which variability in real OTU data is because of noise rather than “signal” of interactions. Here, we simply wanted to verify that small amounts of noise do not fundamentally alter the structural identifiability results and to highlight the structural identifiability results in a toy system.

- If I understand well, the aim of Figure 4 is to illustrate the possibility of estimating parameters from different kind of abundances. It is therefore a key figure of the paper. Is the fit shown actually good? Is there a way to quantify the fit quality? What about the estimation errors? Is the figure actually demonstrating that estimates are correct on the relative scale? It is really not obvious and should be improved/clarified.

Again, we reworked this section for clarity. Our intention with the local practical identifiability was to verify that there is in fact a local maximum of the likelihood surface near the true parameter values. We clarified that the fit is good in the sense that most parameters are estimated to within 10-20% of their true value. Because we use the true initial population size, parameters are estimated exactly (not on a relative scale). We have tried to clarify this in the text.

- In the paper, only two different sets of parameters are assessed. Such a study would be really useful if a more global study was carried out, simulating data on a range of parameters set over realistic parameters intervals, and estimating associated parameters. In particular, the sensitivity and specificity at detecting significant interactions (in the correct direction) would be an interesting measure of efficacy.

We completely agree that a global analysis of a parameter estimation method that includes both process and measurement error in simulated large communities with varying parameter values and levels of error would be extremely interesting and useful. We do not claim to have a method to fit the relative abundance equations globally for large systems. We now state this in the discussion in the paragraph beginning on line 211. Rather, we feel that our structural identifiability results themselves offer insight into the limits of what is possible to estimate with fitting schemes that could be developed in the future.

- How does stochasticity in the simulation model affect estimates? In microbiota studies, usually a limited number of points is available (5, 10, 20?). How does this number affect the reliability of the estimates? What is the minimum number of points required? This should be discussed.

Again, we completely agree that results of any parameter estimation scheme will depend on the quality and quantity of data (as for any systems, even simple linear regression). We touch on this in the new paragraph starting on line 57 and in the Discussion paragraph beginning on line 223. Because we do not have a method for fitting realistic large systems in the absence of prior information regarding parameter

values, we feel that such an analysis (while incredibly useful and important for the future) is beyond the scope of the current paper.

Minor comments

- *Abstract: the abstract could be improved. In particular, the sentence “relative abundance data alone do contain information on relative interaction strengths...” is nearly repeated 6 lines below.*

We agree and deleted the redundancy in the abstract.

- *First equation is not numbered*

Fixed.

- *Line 107, analyses (...) uncover*

Fixed.

- *Figure 4: tracing the 0 line would be useful for the reader.*

Fixed.

Appendix B

Feb 8, 2021

Dear Drs. Drasdo and Chaplain,

Thank you for the encouraging response to our manuscript, "Parameter Identifiability of the Generalized Lotka-Volterra Model for Microbiome Studies." While both reviewers felt that the work constituted a noteworthy contribution, both reviewers felt that several aspects of the manuscript needed further explanation. In particular, we have clarified the methodology and results of the "Local Practical Identifiability in a Synthetic Community" section and modified the discussion for readability and to highlight other potential pitfalls of fitting dynamic models such as the gLV to microbiome time-series data. In addition to these larger changes, we addressed the other more minor comments made by the reviewers. Details of these changes are as follows:

Reviewer: 1

Compared to the first version of the manuscript, the authors reworked all mentioned points so that readability improved significantly. At least from my point of view, it much easier to follow the author's intentions.

Thank you again for your helpful comments. We agree that readability has improved.

However, the section on the analysis of 'local practical identifiability' is rather an example of such a gLV model with relative data that a broad analysis of local practical identifiability. Here, the authors generate data of one noise realisation for simulated measurements and apply a particle MCMC search to fit the model to the simulated data. The fit to the "Value 2" scenario shows obvious deviations from the trajectory of the underlying truth.

This is a good observation, and we agree that there are indeed clear deviations between the model fits and the simulated data. This is to be expected because the simulated data contains both process and measurement error. We have clarified this point in the text on lines 186-188.

Further, they show boxplots of the parameter values from the best MCMC runs to conclude that the likelihood has a unique maximum in the local neighborhood. In my eyes, only vague conclusions about the (local) practical identifiability can be drawn from this boxplot analysis. The box plots from the 'value 2' scenario are quite broad and rather support a non-unique maximum of the likelihood, i.e. a practical non-identifiability. At least histograms of the parameter values and/or their actual corresponding likelihood values are needed to draw conclusions about the mono-modality of the distributions and thus of the uniqueness of the likelihood's maximum. I do not doubt that these models and parameters may be (locally) practical identifiable, but in my eyes the presented analyzes are not strong enough to support this.

We have added the posterior distributions of these parameter values to the Supporting Information R Markdown file, and they generally show monomodality. We now refer to the Supporting R Markdown file on line 186.

Minor points:

- l.99 and other appearances: only 'Saccomani' is the last name of the author in the referenced work, Pia might be the middle name

Fixed.

- l. 129ff : There is something like a sentence missing between equation (6) and equations (7). Where does equation (7) follow from?

There is a sentence between equations 6 and 7: "Substituting equation (6) and its derivative into the N' equation in equation (5) and collecting terms yields..." Essentially, this allows the model to be written as a single equation entirely in terms of x and its derivatives.

- equations (7): x(t) should be simply x

Fixed. Thank you.

- l. 133/134 I still find it difficult to follow the authors how they extract the coefficients from these lines from equation (7)

The list of coefficients is easily extracted in Mathematica using the CoefficientList function. The full code for the structural identifiability analysis is supplied in the Supplemental Mathematica Notebook. These coefficients could also be extracted by hand though the algebra and calculus to get equation (7) becomes cumbersome quickly. As an example, we can see the first three coefficients in equation (7) are:

$$(b_{21} - b_{22}) - (b_{11} - b_{12})$$

$$(b_{22} - 2b_{22}) - (b_{11} - 2b_{12})$$

$$(b_{22} - b_{12})$$

where b_{ij} is $\beta_{i,j}$.

- l. 142 and 149: same sentence used

Fixed.

- l. 151 and 164: convergence of what?

We have clarified that we were interested in whether the likelihood surface has a maximum near the true parameter values on line 151. We have eliminated "convergence" from the text on line 164 as we agree that it was redundant and confusing.

- l. 167: There is no parameter N_0 in Table 1. Also, it remains unclear if parameter value set 1 or 2 is used as reference for scenario 3. Would it be possible to simply expand table 1 by a column with the parameter set used for scenario 3?

We have added the parameter N_0 to Table 1 and clarified how the simulated data were generated on lines 160-161. Specifically, a time series was generated using the parameters Value 2 in table 1, and then parameters estimated using three sets of initial conditions (the two in table 1 and a third where $N(0)=1$).

- l. 184 only the stated simultaneous combination of both error sources is analyzed in this work, not their single contributions.

Fixed.

- l. 212-223 could be shortened

We have shortened the text. Specifically, we cut the paragraph added key information from the paragraph as new text in the paragraph beginning on line 214.

Reviewer: 2

I thank the authors for their point by point answers and updated version of the manuscript which I think is really improved. Figures changes are also notable and definitely improve clarity. Again, I believe this is an important topic and very useful work.

Thank you again for your helpful comments. We agree that clarity has improved.

However, I would like to stress a remaining aspect for which I did not find a proper answer in the revised version and strongly believe this is a key point for the scientific community. Following the comment regarding a larger analysis over the parameter space, the authors replied that they only aimed here at investigating likelihood estimation locally around the true known parameters. Despite I understand that noisy data can generate a lack of identifiability within a small interval around the true value, and that investigating such bias is important; of more importance is, I think, the fact that noisy data and availability of relative counts may lead to select completely different sets of parameters combinations. When such a combination of interactions generates a good likelihood, it may lead to another type of bias, of I think much bigger impact. This latter is, to my mind, a very important point that should not be eluded. This is in particular a key question for those who analyze in practice such systems and data.

This is a very good point. In real systems which are typically large, noisy, and sparsely sampled over time, multiple fundamentally different sets of parameters may lead to good fits---or worse, an entirely incorrect set of parameters may lead to a better fit than the “true” parameters. While we completely agree that is a real and important issue, the specifics of when this occurs likely depends critically on numerous factors specific to a given system (e.g., the interaction network, perturbations/initial conditions, amount of noise, number of samples). It is difficult to see how generalities can be drawn. In contrast, the structural identifiability result is general and provides information on what parameters one can hope to identify. We have added text in the paragraph beginning on lines 226-229 that highlight this important issue.

If this mentioned global analysis of the parameter space is not included in the present paper, I suggest the title to be modified to indicate that the article focuses on the question of bias and identifiability locally.

We have modified the title to clarify that the paper focuses on structural identifiability.